# WEAK SUPERVISION FOR TIME SERIES: WEARABLE SENSOR CLASSIFICATION WITH LIMITED LABELED DATA

## ABSTRACT

Using modern deep learning models to make predictions on time series data from wearable sensors generally requires large amounts of labeled data. However, labeling these large datasets can be both cumbersome and costly. In this paper, we apply weak supervision to time series data, and programmatically label a dataset from sensors worn by patients with Parkinson's. We then built a LSTM model that predicts when these patients exhibit clinically relevant freezing behavior (inability to make effective forward stepping). We show that (1) when our model is trained using patient-specific data (prior sensor sessions), we come within 9% AUROC of a model trained using hand-labeled data and (2) when we assume no prior observations of subjects, our weakly supervised model matched performance with hand-labeled data. These results demonstrate that weak supervision may help reduce the need to painstakingly hand label time series training data.

## 1 INTRODUCTION

Time series data generated by wearable sensors are an increasingly common source of biomedical data. With their ability to monitor events in non-laboratory conditions, sensors offer new insights into human health across a diverse range of applications, including continuous glucose monitoring (Cappon et al., 2017), atrial fibrillation detection (Tison et al., 2018), fall detection (Casilari et al., 2017), and general human movement monitoring (Kumari et al., 2017).

Supervised machine learning with sensor time series data can help automate many of these monitoring tasks and enable medical professionals make more informed decisions. However, developing these supervised models is challenging due to the cost and difficultly in obtaining labeled training data, especially in settings with considerable inter-subject variability, as is common in human movement research (Halilaj et al., 2018). Traditionally, medical professionals must hand label events observed in controlled laboratory settings. When the events of interest are rare this process is time consuming, expensive, and does not scale to the sizes needed to train robust machine learning models. Thus there is a need to efficiently label the large amounts of data that machine learning algorithms require for time series tasks.

In this work, we explore *weakly supervised* (Ratner et al., 2016) models for time series classification. Instead of using manually labeled training data, weak supervision encodes domain insights into the form of heuristic *labeling functions*, which are used to create large, probabilistically labeled training sets. This method is especially useful for time series classification, where the sheer number of data points makes manual labeling difficult.

As a motivating test case, we focus on training a deep learning model to classify freezing behaviors in people with Parkinson's disease. We hypothesize that by encoding biomechanical knowledge about human movement and Parkinson's (Halilaj et al., 2018) into our weakly supervised model, we can reduce the need for large amounts of hand labeled data and achieve similar performance to fully supervised models for classifying freezing behavior. We focus on two typical clinical use cases when making predictions for a patient: (1) where we have no prior observations of the patient, and (2) where we have at least one observation of the patient.

## 2 BACKGROUND

### 2.1 WEAK SUPERVISION

In weak supervision, noisy training labels are programmatically generated for unlabeled data using several heuristic labeling functions which encode specific domain knowledge. These labeling functions are modeled as a generative process which allows us to denoise the labels by learning their correlation structure and accuracies (Ratner et al., 2016). These labeling functions are of the form $\lambda : \mathcal{X} \to \mathcal{Y} \cup \emptyset$ which take in a single candidate $x \in \mathcal{X}$, and output a label $y \in \mathcal{Y}$ or $\emptyset$, if the function abstains. Using $n$ labeling functions on $m$ unlabeled data points, we create a label matrix $L = (\mathcal{Y} \cup \emptyset)^{m \times n}$. We then create a generative model from this label matrix and three factor types (labeling propensity, accuracy, and pairwise correlation) of labeling functions:

$$\phi_{i,j}^{Lab}(L, Y) = \mathbb{1}\{L_{i,j} \neq \emptyset\}$$

$$\phi_{i,j}^{Acc}(L, Y) = \mathbb{1}\{L_{i,j} = y_i\}$$

$$\phi_{i,j,k}^{Corr}(L, Y) = \mathbb{1}\{L_{i,j} = L_{i,k}\} \quad (j, k) \in C$$

where C are the potential correlations. Next, we concatenate all these factors for a given data point $x_i$ and all labeling functions $j = 1...n$, resulting in $\phi_i(L, Y)$, and learn the parameters $w \in \mathbb{R}^{2n+|C|}$ to maximize the objective:

$$p_w(L, Y) = Z_w^{-1} \exp\left(\sum_{i=1}^{m} w^T \phi_i(L, y_i)\right)$$

With this generative model, we can then generate probabilistic training labels, $\tilde{Y} = p_{\hat{w}}(Y|L)$ where $\hat{w}$ are the learned parameters in the label model.

Using these probabilistic labels, we can train a discriminative model that we aim to generalize beyond the information encoded in the labeling functions. We do this by minimizing the expected loss with respect to $\hat{Y}$:

$$\hat{\theta} = \arg\min_{\theta} \sum_{i=1}^{m} \mathbb{E}_{y \sim \tilde{Y}} l(h_\theta(x_i), y)$$

As we increase the amount of unlabeled data, we increase predictive performance (Ratner et al., 2017).

## 3 METHODOLOGY

### 3.1 DATASET

We use a dataset that contains series of measurements from 36 trials from 9 patients that have Parkinson's Disease (PD) and exhibit freezing behavior. PD is a neurodegenerative disease marked by tremor, loss of balance, and other motor impairments, that affects over 10 million people worldwide. Freezing of gait (FOG) — a sudden and brief episode where an individual is unable to produce effective forward stepping (Giladi et al., 1992; 1997) — is one of the disabling problems caused by PD, and often leads to falls (Bloem et al., 2004).

In this dataset, subjects walked in a laboratory setting that the investigators designed to elicit freezing events. Leg or shank angular velocity was measured during the forward walking task using wearable inertial measurement units (sampled at 128 Hz), which were positioned in a standardized manner for all subjects and tasks on the top of the feet, on both shanks, on the lumbar, and chest trunk regions.

### 3.2 PREPROCESSING

In this work, we focus on sensor streams from both shanks, though it is straightforward to include the other sensor streams (e.g. lumbar, feet, etc.). From each Turning and Barrier Course run, we extract left and right ankle gyroscope data in the z-direction from each trial (up to 4 trials per course run), along with the gold labels for these trials, which were manually recorded by a neurologist. We

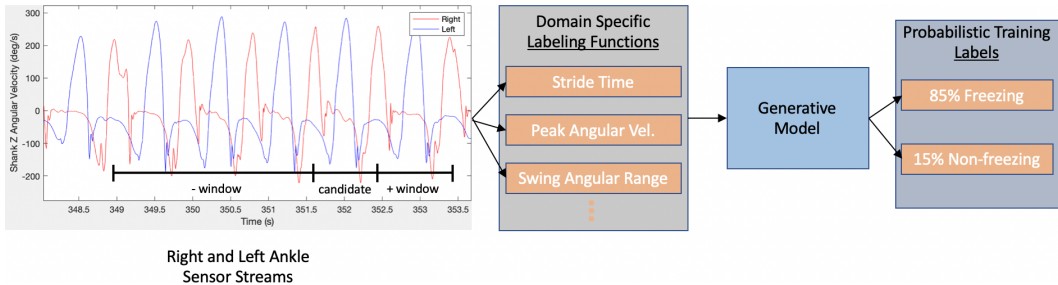

Figure 1: Example: From Raw Sensor Data to Probabilistic Labels

Table 1: Labeling Function Evaluation Results

| Labeling Function | Coverage (%) | Emp. Accuracy (%) | F1 |
|---|---|---|---|
| **Swing Angular Range** | 71.2 | 55.3 | 64.3 |
| **Angular Range Average** | 71.9 | 59.6 | 60.5 |
| **Peak Angular Shank Velocity** | 42.7 | 78.1 | 75.5 |
| **Stride Time** | 48.9 | 53.9 | 56.2 |
| **Variance in Angular Shank Velocity** | 75.4 | 63.2 | 65.0 |

combine the data from all trials from all course runs, and segment the sensor data by gait cycle (of the right leg) which is computed analytically from the angular velocity of ankle sensor data. In this case, we define gait cycle as the time period between two successive peaks on an angular velocity versus time plot.

We then define a single candidate to be $x^{a \times b} \in \mathcal{X}$ where $a$ is the number of sensor streams and $b$ is the sequence length. For our task, $a = 2$ since we use the left and right ankle sensor streams, and $b$ is the sequence length for a single gait cycle (which slightly varies from cycle to cycle).

## 4 EXPERIMENTAL EVALUATION

### 4.1 LABELING FUNCTIONS

To programatically label data, we use five labeling functions which draw on domain specific knowledge and empirical observations. Specifically, these labeling functions target features which can distinguish freezing and non-freezing events. For all labeling functions, we assign positive, negative, or abstain labels based on empirically measured threshold values from the validation set. For example, one heuristic we employ uses stride time arrhythmicity (Plotnik et al., 2005; 2007), which we calculate as average coefficient of variation for the past 3 stride times of the left and right leg. For this function, we label a candidate as *freezing* if the arrhythmicity of that candidate is greater than 0.55, and *not freezing* if the arrhythmicity is less than 0.15. If arrhythmicity for a particular candidate is in between these two values, we abstain. Other labeling functions we use involve the swing angular range of the shank, and the amplitude and variance in shank angular velocity.

Using these labeling functions, we build a generative label model and predict probabilistic labels $y \in \mathcal{Y}$ for each candidate $x^{a \times b} \in \mathcal{X}$ in the training set (Figure 1). See Table 1 for the individual performance of each labeling function.

### 4.2 END MODEL

We then train a discriminative model on the probabilistic labels from the generative model that incorporates the labeling functions discussed in the last section. We use a single layer bi-directional LSTM and hidden state dimension 300 for our end model that takes in a multivariate sensor stream as input. Since we use time series data from only the left and right ankle sensors, our input is two dimensional. In order to provide longer temporal context, we pass in a windowed version of each

Table 2: Performance in Each Setting

|  | Session Splits | | Patient Splits | |
|---|---|---|---|---|
| Supervision Type | F1 | AUROC | Mean F1 (SD) | Mean AUROC (SD) |
| Hand-labeled | 70.3 | 85.2 | 35.2 (31.1) | 69.9 (14.8) |
| Weakly Supervised | 60.7 | 77.3 | 35.0 (29.3) | 71.2 (11.24) |

Table 3: Metrics for Individual Patients (hand-labeled supervision)

| Metrics | P1 | P2 | P3 | P4 | P5 | P6 | P7 | P8 |
|---|---|---|---|---|---|---|---|---|
| Positive Class Percentage | 4.07 | 6.53 | 38.2 | 60.3 | 36.1 | 15.8 | 35.7 | 11.2 |
| F1 | 0 | 26.8 | 69.3 | 78.4 | 67.7 | 37.3 | 0 | 2.15 |
| AUROC | 42.9 | 80.0 | 84.7 | 74.6 | 82.6 | 80.7 | 62.0 | 51.4 |

candidate that includes the last three gait cycles and the next gait cycle (Figure 1). Since sequence length of a single gait cycle slightly varies, we then pad these sequences and truncate any sequences over a pre-defined maximum sequence length. To provide more contextual signal, we also add multiplicative attention to pool over the hidden states in the LSTM.

### 4.3 RESULTS

We evaluate our weakly supervised model in two typical clinical settings, and compare performance with that of a fully supervised model. In the first setting, we split the data into training/validation/testing by trials/sessions. In this setting, both the validation and testing set have a single trial from each patient, and the training set has one or more trials from each patient. In the second setting, we split data by patient. In this case, the testing (and validation) set contains all the trials from a novel patient. We then cross-validate on each patient. These results are summarized in Table 2.

From the session splits setting, we note that our weakly supervised model comes within 10 points in F1 score and 8 points in AUROC of the fully supervised (hand-labeled) model. In the patient splits setting, our weakly supervised model matches the performance of the fully supervised model.

In both supervision types, it is clear that our end to end system performs significantly better when it has seen 1 or more sessions of a particular patient before. In the patient splits setting, our system has difficulty generalizing to certain patients. For example, in Table 3 we see that the fully supervised model has trouble predicting freezing events for patients P1, P7, and P8 in particular. These difficulties are inherent to the problem — each patient exhibits different freezing behaviors, and some, such as P1, P2, and P8, have relatively rare freezing events. This highlights why, at least for this task, it is critical to have as many subjects as possible in the dataset — an objective that is far easier to meet if hand labels are not required.

## 5 CONCLUSION AND NEXT STEPS

Our work demonstrates the potential of weak supervision on time series tasks. In both experiments, our weakly supervised models performed close to or match the fully supervised models. Further, the amount of data available for the weak supervision task was fairly small — with more unlabeled data, we expect to be able to improve performance (Ratner et al., 2017). These results show that costly and time-intensive hand labeling may not be required to get the desired performance of a given classifier.

In the future, we plan to add more and different types of sensor streams and modalities (e.g., video). We also plan to use labeling functions to better model the temporal correlation between individual segments of these streams, which can potentially improve our generative model and hence end to end performance.

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
