# OpenReview forum: "Weak Supervision for Time Series: Wearable Sensor Classification with Limited Labeled Data"
_ICLR.cc/2019/Workshop/LLD — Submitted to LLD 2019_

### Official Review · AnonReviewer1 · 2019-04-06
**Interesting application, despite lacking important details**

**Rating:** 4
**Confidence:** 1

**Review:**

This paper proposes a two-tier machine learning system, combining a generative model and a discriminative model. The discriminative model is an LSTM, and the generative model is a "label model", which learns a labeling function based on domain-specific knowledge and empirical observations. The authors report successful experiments on wearable sensor data, which is an interesting and under-studied application domain of representation learning.

The introduction and discussion of prior literature is well written. However, the paper lacks a precise description of its main contribution, and how this contribution sets it apart from prior literature.

My other piece of criticism is that the description of the entire system is not summarized by a single diagram or algorithm listing. I had to jump back and forth between pages 2 and 3 to understand what the paper was about. Furthermore, calling the label model a generative model may be confusing to the reader. This choice of terminology may perhaps be appropriate, but in any case, the authors should strive for the maximum level of clarity. In the case of wearable sensor data, it is not clear what the end goal of the LSTM is: is it to predict freezing? Likewise, it is not clear how the label model manages to approximate domain-specific labeling functions.

---

### Official Review · AnonReviewer2 · 2019-04-07
**Experimental weaknesses and not novel enough**

**Rating:** 1
**Confidence:** 2

**Review:**


* Paper summary:

This paper proposes to apply weak supervision to a wearable time-series classification problem.

Weak supervision consists in using a collection of heuristic labeling functions to build a generative function of labels. A classifier is then trained using labels generated by this probabilistic model. This is definitely appealing when no human labeler is available.

This method is applied to a time-series classification problem relying on sensor data, with a bidirectional LSTM model. Depending on the considered setting, the weakly-supervised model achieves a performance which is significantly lower or slightly higher than a strongly-supervised model (trained with the true labels).

* Decision
This paper suffers from experimental weaknesses and is not novel enough in terms of ideas to reach acceptance.

* General remarks:
- Major experimental weakness: the authors only compare the weakly supervised model to the strongly supervised one and to the *individual* labeling functions (which perform quite poorly individually). However, due to the very construction of weak supervision, a probabilistic *ensemble model* is constructed during weak supervision. The performance of this simple ensemble of heuristic functions is not reported by the authors. Thereby, it is impossible to distinguish the contribution of the weak supervision from the simple ensembling of heuristic functions, without any learning. The claim of the authors is therefore not supported by their experimental apparatus.
- The paper is quite well written and easy to follow at high level, despite some imprecisions.
- The methodology, task and dataset are not presented rigorously enough in section 3. (Cf detailed remarks)


* Detailed questions
1. What is exactly the classification task to be performed? For instance, in 3.2, x is first defined to have the length of a single gait cycle, but is then extended with windows before and after it.
2. What is the total number of samples? The authors claim in the conclusion that it is « small ». A back of the envelope computation given numbers in Figure 1 leads me to think that each sample consists of a 2s window, plus ~3 s around it, so 5s in total. Sampling rate is 128Hz and there are two channels, so each channel has dimension ~1300. The recording shown in Figure1 goes until 350s, so 350/5~70 samples / recording, with 9 patients and 36 trials per patient this leads to ~2300 recordings. This is not a « small data » problem.
3. What value of the correlations C is used in the actual labeling functions?
4. What kind of padding is used for the LSTM model?

---

### Decision · Program_Chairs · 2019-04-16
**Acceptance Decision**

**Decision:**

Reject

**Comment:**

Reviewers found issues with novelty and clarity